# The *3′UTR VNTR SLC6A3* Genetic Variant and Major Depressive Disorder: A Systematic Review

**DOI:** 10.3390/biomedicines11082270

**Published:** 2023-08-15

**Authors:** Bruna Rodrigues Gontijo, Isabella Possatti, Caroline Ferreira Fratelli, Alexandre Sampaio Rodrigues Pereira, Larissa Sousa Silva Bonasser, Calliandra Maria de Souza Silva, Izabel Cristina Rodrigues da Silva

**Affiliations:** 1Graduate Program in Health Sciences and Technologies, Faculty of Ceilandia, University of Brasilia, Federal District, Brasilia 72220-900, Brazil; brunargontijo.unb@gmail.com (B.R.G.); isabellapossatti@gmail.com (I.P.); carolfratelli@gmail.com (C.F.F.); prof.alexandresampaio@gmail.com (A.S.R.P.); cdssilva@gmail.com (C.M.d.S.S.); 2Graduate Program in Health Sciences, University of Brasilia, Federal District, Brasilia 70910-900, Brazil; laribonasser@gmail.com

**Keywords:** *SLC6A3*, DAT1, rs283631170, untranslated region, Major Depressive Disorder, genetic polymorphism

## Abstract

Major Depressive Disorder (MDD) is a disabling and particularly persistent mental disorder that is considered to be a priority public health problem. The active human dopamine transporter (DAT), which is encoded by the *SLC6A3* gene, regulates the dopamine concentration in the synaptic cleft. In this sense, this neurotransmitter is primordial in modulating human emotions. This systematic review aims to verify the *SLC6A3 (DAT1) 3′UTR VNTR (rs28363170)* gene variant’s SS (9R/9R) genotype and S (9R) allele frequency fluctuation and its influence on the modulation of pharmacotherapy in MDD. For this purpose, we searched different databases, and after applying the eligibility criteria, six articles were selected. Studies have shown an association between the SS (9R/9R) genotypic and S (9R) allelic presence with the risk of developing MDD, in addition to influencing the decrease in response to antidepressant therapy. However, despite the findings, disagreements were observed between other studies. For this reason, further studies with the *SLC6A3 3′UTR VNTR* (*rs28363170*) variant in different populations are necessary to understand this polymorphism’s role in the onset of this disease.

## 1. Introduction

Major Depressive Disorder (MDD) is a condition responsible for physical and psychological changes in the individual and causes significant disability in society. Among these changes is the onset of symptoms such as depressed mood, loss of interest or pleasure in daily activities, and intense guilt [1,2]. This disease is also characterized by reduced affection, cognitive dysfunction, and significant psychosocial impairment that persists for extended periods [3].

Over time, this disease and its symptomatology have had an increasingly more prominent global prevalence [4]. This pathology is almost twice more frequent in females than in males, and in both sexes, its prevalence peak occurs between 20 and 30 years, with a more modest peak between 50 and 60 years [5]. According to the World Health Organization (WHO) [6], the total number of individuals suffering from depressive symptoms increased by 18.4% between 2005 and 2015. The COVID-19 pandemic’s emergence also negatively influenced mental health, further increasing its prevalence and incidence, and brought tremendous economic and social consequences over time. Places hit hardest by the pandemic witnessed the most substantial increases in prevalence rates for MDD and anxiety disorders [7].

Several social, cultural, biological, and genetic factors can modify a gene’s expression through epigenetic mechanisms. Although MDD’s etiology remains unknown, these changes may have a preponderant role in its development [5,8]. Among several theories, the monoaminergic hypothesis can demonstrate MDD’s pathogenesis. For instance, MDD patients’ central nervous system negatively regulates the levels of neurotransmitters such as serotonin (5-HT), norepinephrine (NE), and dopamine (DA) by reducing their activation [5]. DA functions in the brain’s reward, concentration, motivation, and psychomotor speed systems and in the brain’s ability to experience pleasure, possibly modulating human emotions [9]. Consequently, dopaminergic neurotransmission genes might be associated with personality trait development and mental illnesses [10].

The dopamine transporter (DAT1) regulates and controls dopamine levels in the brain and among dopaminergic neurons in the synaptic cleft through the reuptake of DA in the presynaptic terminals [11]. The *DAT1* gene, also called *SLC6A3*, is located on chromosome 5p15.3 [12,13]. One of its most described polymorphisms is the 40 bp *VNTR* in the *3′UTR*. This variant, in turn, has seven known alleles originating from it, ranging from 3 (3R) to 11 (11R) tandem repeats, with the best known being 9 (9R) or 10 (10R) tandem repeats [14] (Figure 1). Genotypically, the homozygosity of the 9R (S) allele and the 10R (L) allele correlates with transcriptional differences [14] (Figure 2). The 9R/9R (SS) genotype reduces DAT expression, causing an increase in dopamine levels in the synapse and, consequently, greater ventral striatal reactivity. By contrast the 10R (LL) homozygote provides a higher DAT concentration, better DA reuptake, and, consequently, reduces this neurotransmitter’s synaptic availability, possibly decreasing depressive symptoms [10,14,15].

Thus, this systematic review aims to evaluate the *SLC6A3 (DAT1) 3′UTR VNTR* (*rs28363170*) genetic variant influence on MDD, as such information might help identify risk factors to assist in new forms of diagnosis, treatment, and ways to improve the quality of life of patients with psychopathology. For this, we examined the *3′UTR VNTR* variant’s SS (9R/9R)/S(9R) frequency fluctuations and their associations with MDD in different populations.

## 2. Methods

### 2.1. Search Strategy and Selection Criteria

This systematic review followed the PRISMA guidelines for systematic reviews and meta-analyses and is registered in the Prospective of Systematic Reviews (PROSPERO) under CRD42022320374.

The inclusion criteria were based on the following aspects: population, exposure, comparison, outcome, and study type (PECOS). (1) Population: research participants with Major Depressive Disorder (MDD). (2) Exposure: *SLC6A3 3′UTR VNTR* gene variant (rs28363170). (3) Comparison: the *3′UTR VNTR* genetic variant’s SS (9R/9R) genotypic frequency, S (9R) allelic frequency, or both. (4) Outcome: the *3′UTR VNTR* genetic variant’s SS (9R/9R)/S (9R) frequency fluctuation in different populations. (5) Study type: observational and intervention.

For this purpose, we included original, observational, or interventionist studies, in English or Portuguese, describing the impacts of the *SLC6A3 3′UTR VNTR* genetic variant’s SS (9R/9R) genotype/S (9R) allele on MDD in humans and that also presented the laboratory and statistical methods used. However, articles that presented incomplete data, such as methodology or statistical data, systematic reviews or meta-analyses, and abstracts, were excluded. For the development of this study, the alleles of the *SLC6A3 3′UTR VNTR* gene were considered in short S (9 repetitions) and Long L (10 repetitions) alleles.

The research instruments used for the bibliographic survey, conducted in March 2023, were the following databases: PubMed, Web of Science, Virtual Health Library (BVS), and Scopus. The indexed terms that were searched reflected this review’s determined exposure and outcome and were based on the Medical Subject Headings (MeSH) vocabulary and The ALLele FREquency Database (ALFRED), developed by the Yale Center for Medical Informatics, which catalogs allele frequency data for different polymorphisms in human populations for free. The ALFRED database can be accessed via the website https://alfred.med.yale.edu/alfred/index.asp [16].

Accordingly, the indexed terms (descriptors) were as follows: (DAT1 OR ADD OR SLC6A3 OR “dopamine transporter1”) AND (polymorphism, genetic OR “tandem repeat” OR “untranslated region”) AND (Depressive Disorder, Major). The descriptors DAT1, ADD, SLC6A3, and dopamine transporter 1 were found on the ALLele FREquency Database (ALFRED), and the others were found on Medical Subject Headings (MeSH).

### 2.2. Study Selection and Data Extraction

Two reviewers (BR and IP) selected articles and then extracted the data in two stages. In the first stage, each reviewer independently examined each article’s title and abstract, investigating its eligibility according to the PECOS inclusion criteria. For this step, the Qatar Computing Research Institute (QCRI)’s Rayyan tool was employed to analyze and remove duplicate articles. In the second stage, the same two reviewers (BR and IP) independently analyzed the full text of articles that initially passed the first stage with the help of the Mendeley Desktop software (version 1.19.4). They extracted the following information: author, study’s title, objective, year of publication, country of the study’s origin, sample size, laboratory methodology, results, statistical values (*p*-value), and SS (9R/9R)/S (9R) frequency. These extracted data were placed in a table format, using the Microsoft Office Professional Plus Platform version 2021. In the case of disagreement at any of the steps, a third reviewer (IS) was consulted.

### 2.3. Bias Risk of Each Study

The risk of study bias was analyzed using the Genetic Risk Prediction Studies (GRIPS) guideline. Composed of 25 items, this guideline aims to offer a guide for how these types of studies can achieve quality and completeness, enhancing their reporting transparency and improving the synthesis and application of information from multiple studies that might differ in design, conduct, or analysis [17]. In this systematic review, two reviewers (BR and IP) independently analyzed the absence/presence of 20 of these items to assess the quality of the selected studies’ methodology, results, and discussion. A third reviewer (IS) was consulted the in case of any disagreement. The presence of at least 75% of these items classified the article as “good quality”.

## 3. Results

During our research, we identified a total of 186 scientific articles. After removing duplicates and verifying articles, we selected 115 titles and abstracts to be analyzed by following the aspects delimited in our PECOS strategy. Applying pre-established inclusion and exclusion criteria yielded the six articles analyzed in this systematic review (Figure 3).

Figure 4 presents the selected articles’ continents of origin, in which 50% are from the European continent, 33.3% from the Asian continent, and 16.7% from the North American continent.

Analyzing the selected articles’ sample population (Table 1), we noted that all presented a higher frequency of females, except for Frisch et al.’s [18] study, in which the distribution by gender was unspecified. Regarding the SLC6A3 3′UTR VNTR gene, the most frequent genotypic frequencies were LL and LS, and the least was SS. All research participants in the selected articles were over 18 years old.

## 4. Discussion

### 4.1. SLC6A3 3′UTR VNTR (rs28363170) Variant and Its Genotypic Frequency in Major Depressive Disorder (MDD)

Major Depressive Disorder (MDD) can be caused by a dysfunction of the dopaminergic reward network that connects the ventral striatum to the orbitofrontal and medial prefrontal cortices [22,23,24,25]. One of the mechanisms responsible for regulating the dopaminergic system is the dopamine transporters (DATs), which, together with other mechanisms, help modulate extracellular striatal dopamine (DA) concentrations in the synaptic cleft in different brain regions, mainly the cortical and subcortical regions [26]. Located on the membrane of presynaptic terminals, DAT is responsible for regulating the intensity and duration of dopaminergic transmission in the synaptic cleft and the reuptake of dopamine from presynaptic cells in the striatum and midbrain, in addition to interfering with the clearance of dopamine in the extracellular striatum region [27]. Reducing dopamine levels diminishes striatal DAT’s density and functionality—a compensatory downregulation feedback mechanism to adjust the reduced DA concentration [27]. The SLC6A3 (DAT1) 3′UTR VNTR (rs28363170) polymorphism alters DAT expression, with SS (9R/9R) genotype carriers tending to be associated with a more severe depression course [22].

Observational and intervention studies have verified the *SLC6A3 (DAT1) 3′UTR VNTR* (*rs28363170*) variant genotypic and allelic frequency to analyze its potential association with MDD (Table 1). According to the *SLC6A3 3′UTR VNTR* genotype frequencies, this systematic review’s selected articles agree on the 9R and 10R alleles’ influence on MDD. Two studies with the same population sample reanalyzed [14,22] indicated a possible link between the SS (9R/9R) genotype and a more severe depression form. Having at least one S (9R) allele was enough to offer a risk of developing MDD [14]. On the other hand, the LL (10R/10R) genotype behaves as a protective factor, as it is associated with a low risk of developing MDD, in addition to improving the response to antidepressants [19,20].

López-León et al.’s [28] meta-analysis reviewed case-control studies that analyzed genetic variants’ associations with MDD; among the twenty evaluated polymorphisms was the SLC6A3 3′UTR VNTR, with three studies—a total of 151 cases and 272 controls. Interestingly, these studies demonstrated a more significant increase in the risk of developing MDD for LS (10R/9R) carriers compared to the LL (10R/10R) genotype carriers (reanalyzed pooled OR = 2.06; CI = 1.25–3.40). However, only one of these studies showed a statistically significant value.

Bieliński et al.’s [10] Polish study investigated the effect of the COMT Val158Met and SLC6A3 3′UTR VNTR polymorphisms on 364 obese research participants with depressive symptoms and a Body Mass Index (BMI) between 30 and over 40 kg/m^2^. They found that individuals with the SLC6A3 3′UTR VNTR’s SS (9R/9R) genotype had a higher Beck Depression Inventory (BDI) depression scale (*p* = 0.022) and Hamilton Depression Rating Scale (HDRS or Ham-D) (*p* = 0.00001) score, in addition to a higher BMI (*p* = 0.001).

Similarly, Rafikova et al. [22] looked for associations between depressive episodes (DEs), recurrent depression (RD), and mixed anxiety–depressive disorder (MA-DD) and different polymorphisms, including SLC6A3 3′UTR VNTR. This polymorphism’s S (9R) allele or alleles with fewer repetitions in the allelic distribution were more common in the case groups than the control; however, this difference was only statistically significant in the MA-DD group (DE—*p* = 0.355; RD—*p* = 0.199; and MA&DD—*p* = 0.005). The SS genotype distribution in the MA-DD group was also statistically significant (*p* = 0.025). Hence, the presence of at least one S allele was associated with the risk of developing MA&DD (OR = 3.93).

In a different study, Rafikova et al. [14] pooled the genotyping results from their previous study for the loci of the dopaminergic, serotonergic, and endocrine systems and investigated their association with the same depressive disorders (DE, RD, and MA-DD) through various inheritance models. The *SLC6A3 3′UTR VNTR* variant correlated with DE in the codominant (*p* = 0.024), dominant (*p* = 0.015), superdominant (*p* = 0.035), and recessive (*p* = 0.011) models. Their genotype frequency analysis showed that the SS and LS carriers were more frequent in individuals with DE, and in contrast, LL (10R/10R) genotype carriers had the lowest risk of developing DE (OR = 0.57, 0.35–0, 95, *p* = 0.032). The correlation between recurrent depression was significant in the dominant (*p* = 0.01) and additive (*p* = 0.015) models. Similarly, LL genotype carriers have a low risk of developing RD (OR = 0.59, 0.37–0.94, *p* = 0.025). The most intense association was between the *SLC6A3 3′UTR VNTR* variant and MA-DD. The SS genotype correlated with a higher risk of developing MA-DD (OR = 3.93, 1.18–13.13, *p* = 0.026), while the LL genotype seemed to play a protective role against it (OR = 0, 48, 0.29–0.8, *p* = 0.005).

Other studies suggest no association between this polymorphism and MDD. Frisch et al. [18] investigated dopamine and serotonin transporters’ involvement in depression pathophysiology by genotyping 102 MDD patients and 172 healthy controls for various polymorphisms, including the *SLC6A3 3′UTR VNTR*. Their results showed no statistical difference between MDD patients and healthy controls regarding the *SLC6A3 3′UTR VNTR* genotypic and allelic distribution (*p* > 0.003).

Analogously, Huang et al.’s [21] case-control study analyzed 17 polymorphisms, including the *SLC6A3 3′UTR VNTR*, to assess personality traits in a Han Chinese population (435 healthy controls and 582 MDD patients). The authors divided MDD patients into two distinct clinical groups, those with a positive family history (186 patients) and those without a family history (396 patients), and found no association between MDD and specific personality traits (*p* = 0.986). Furthermore, neither the haplotype analysis nor the conventional correction correlated the *SLC6A3 3′UTR VNTR* genetic variant with MDD.

MDD is considered a contributing factor for developing suicidal behavior, likely due to neurocognitive changes that increase a person’s vulnerability to this behavior [29]. One such change is in neurotransmitter signaling. For instance, individuals with MDD who attempted self-extermination had reduced striatal dopaminergic signaling [30]. Therefore, new research into the links between suicide and genes coding serotonergic and dopaminergic pathways is essential, particularly because self-extermination is also one of the leading causes of death and disability in individuals with MDD [31,32].

Rafikova et al. [12] verified the contribution of 11 polymorphisms, among them the SLC6A3 3′UTR VNTR, to the suicidal behavior and severity of depression and anxiety symptoms in a Russian population (100 patients with repeated suicide attempts and 154 healthy controls). The *SLC6A3 3′UTR VNTR’s* LL genotype appeared more regularly in controls than in cases and was significantly associated with a lower risk factor for suicide behavior (OR = 0.48; *p* = 0.005). When observing the inheritance model, suicidal behavior or the *SLC6A3 3′UTR VNTR* variant correlated significantly with the codominant (*p* = 0.006), dominant (*p* = 0.001), superdominant (*p* = 0.004), and log-additive (*p* = 0.004) models.

### 4.2. SLC6A3 3′UTR VNTR (rs28363170) Variant and Pharmacotherapy

With the development of genetics, the need arose to research whether specific genes and their variants interfere individually or collectively with medications in either their effectiveness, their adverse effects, or both originating pharmacogenetics [33].

Hellwig et al. [23] investigated whether sleep deprivation and electroconvulsive seizures affect striatal dopamine transporter (DAT) availability. Sixteen MDD patients with MDD and twelve controls participated in the study and underwent imaging and SLC6A3 3′UTR VNTR genotyping. Their results indicated that sleep deprivation significantly reduced striatal DAT availability (*p* < 0.01) and that the S (9R) allele presence boosted this effect.

Lavretsky et al. [20] examined the role of dopamine- and serotonin-related genes’ polymorphisms in the clinical and cognitive characteristics of patients with late-life depression. For this, 15 outpatients with late-life depression were studied in a ten-week double-blind pilot trial, using methylphenidate combined with citalopram or citalopram and placebo alone. Individuals with the *SLC6A3 3′UTR VNTR’s* LL (10R/10R) genotype had more significant executive cognitive dysfunction at baseline (*p* = 0.05). Nonetheless, these individuals responded better to the methylphenidate–citalopram treatment, showing a greater reduction in depression severity over time than other groups (*p* = 0.049).

In contrast, Kirchheiner et al.’s [19] prospective cohort study evaluated the influence of the *SLC6A3 3′UTR VNTR* and the *SLC6A4* insertion/deletion genetic polymorphisms on the modulation of the antidepressant response. The *SLC6A3 3′UTR VNTR* variant seemed to influence the antidepressant therapy response, as SS (9R/9R) genotype carriers responded less rapidly than the other genotypes (*p* = 0.0037). Regarding the study’s pharmacogenetic analysis and considering five groups of most frequently used therapeutic agents (mirtazapine, selective serotonin reuptake inhibitors (SSRIs), tricyclics, venlafaxine, and others), the *SLC6A3 3′UTR VNTR* variant influenced more prominently the response to SSRIs, as individuals with at least one L (10R) allele responded better to the treatment decreasing their Hamilton Rating Scale for Depression (HDRS) scores (*p* = 0.026).

In another pharmacogenetic study, Yin et al. [34] determined SSRIs’ antidepressant responses regarding several genetic polymorphisms. After the 6-week randomized and double-blind trial with 290 MDD patients, they found no statistical differences for the *SLC6A3 3′UTR VNTR* polymorphism between responders and non-responders (*p* > 0.05).

### 4.3. Selected Articles’ Quality Assessment and Limitations

The Genetic Risk Prediction Studies (GRIPS) guideline was employed to assess the quality of this systematic review’s selected association studies (Appendix A). Twenty of the guideline’s twenty-five items were selected to evaluate the particularities of each article’s methods, results, and discussion. Of the six studies evaluated, all had at least 15 of the 20 items evaluated (adequacy % of less than 75%).

All articles [14,18,19,20,21,22] described participants’ eligibility criteria and selection sources and methods. They also had a generalized discussion and, when pertinent, revealed the importance of their study. Nonetheless, only 83.3% of the articles [14,18,20,21,22] discussed the relevance of the study results for health care. Only one article [19] (16.6%) specified how it dealt with missing data in the analysis.

## 5. Conclusions

This systematic review found that a preliminary agreement regarding the *SLC6A3 (DAT1) 3′UTR VNTR (rs28363170)* polymorphism association with MDD is inconclusive. Two similar studies with the same population sample and a small sample study indicate that the S (9R) allele’s presence may be a risk factor for developing MDD and that the L (10R) allele’s presence is a protective factor. In contrast, the two remaining studies indicated that this polymorphism does not influence MDD.

The limitations of this study are that few articles met the eligibility criteria for this review and, consequently, very few populations with different ethnicities were analyzed. Therefore, further studies with a significant sample size are needed to assess the behavior of this genetic variant’s genotypic/allelic frequency distribution in diverse populations and help understand this polymorphism’s role in MDD conclusively.

Understanding the *SLC6A3* gene, its polymorphisms, and its mechanisms’ role in MDD etiology might contribute to the establishment of new, more modern forms of early diagnosis and increase patients’ quality of life.

## Figures and Tables

**Figure 1 biomedicines-11-02270-f001:**
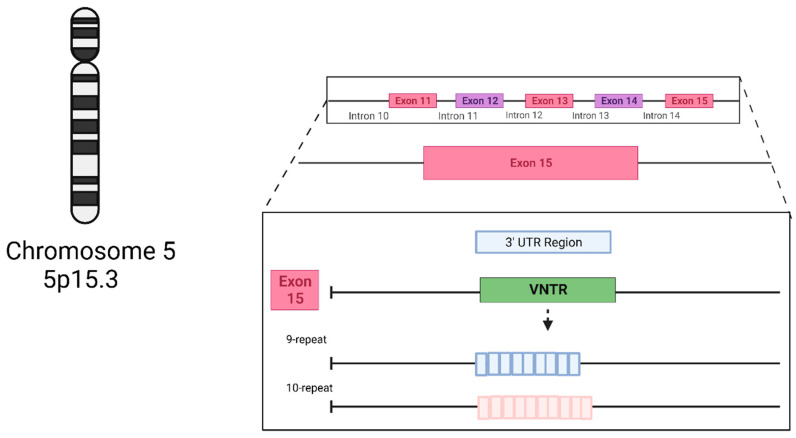
The structure of the *SLC6A3 3′UTR VNTR (rs28363170)* genetic variant’s main alleles. Created with BioRender.com.

**Figure 2 biomedicines-11-02270-f002:**
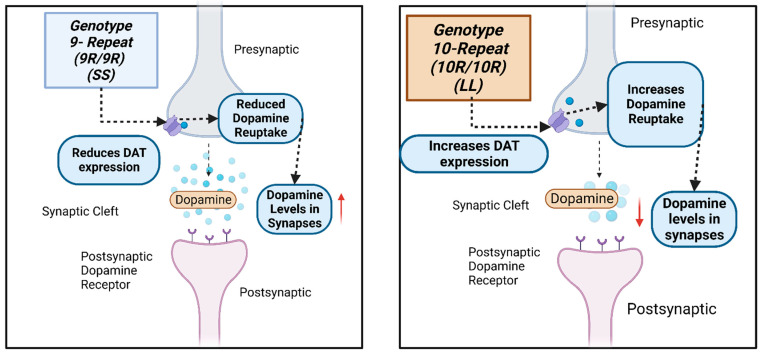
A schematic drawing of the how the *SLC6A3 3′UTR VNTR (rs28363170)* genetic variant influences dopamine levels at the dopaminergic neuron’s postsynaptic and presynaptic regions. Created with BioRender.com.

**Figure 3 biomedicines-11-02270-f003:**
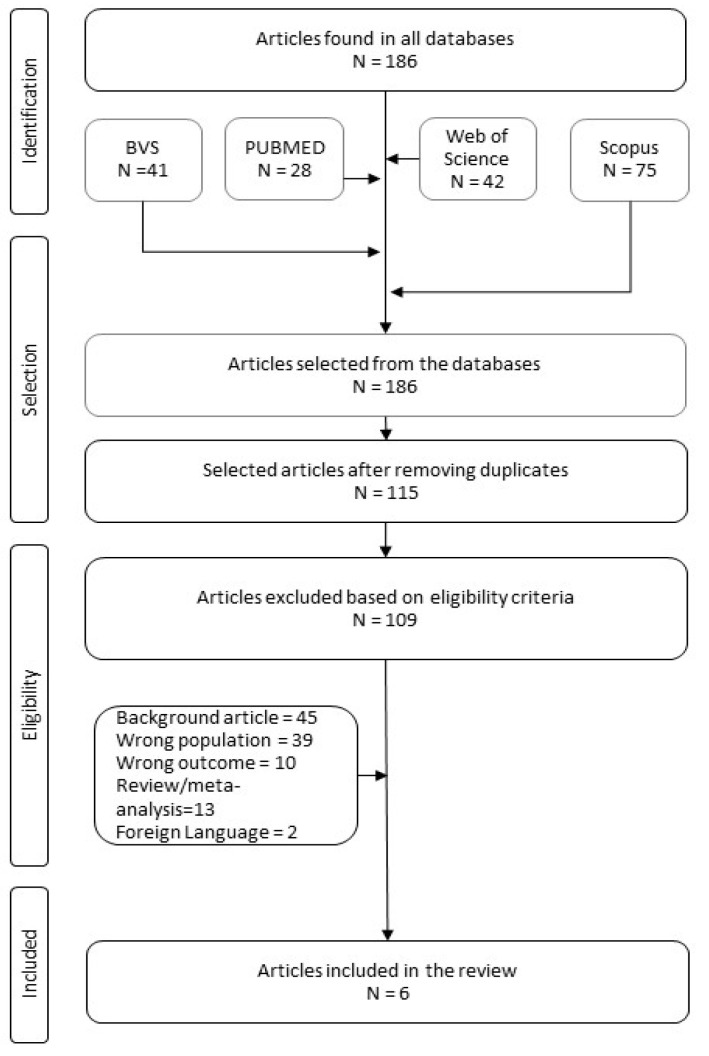
Bibliographic research flowchart.

**Figure 4 biomedicines-11-02270-f004:**
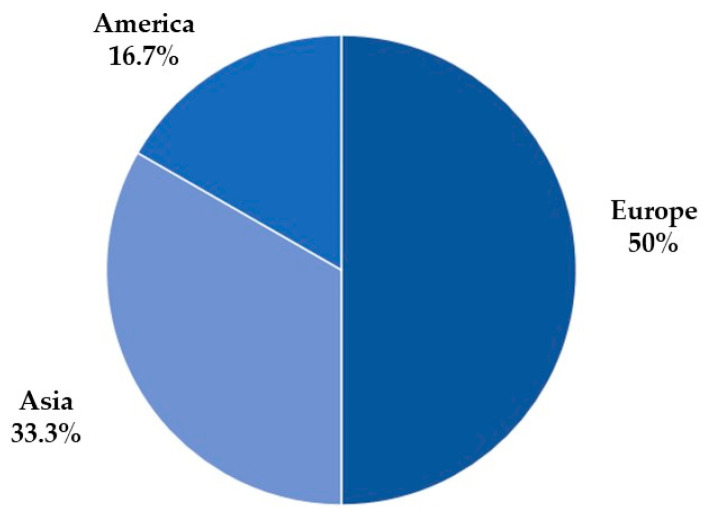
The number of articles by continent.

**Table 1 biomedicines-11-02270-t001:** Comparison of different studies that evaluated the SLC6A3 (DAT1) 3′UTR VNTR (rs28363170) gene variant’s effect on Major Depressive Disorder (MDD).

Author	Title	Objective	Year	Country	Sample (N)	Results	*p*-Value	Genotypic Frequency
Frisch et al. [18]	“Association of unipolar major depressive disorder with genes of the serotonergic and dopaminergic pathways”	Analyze serotonergic and dopaminergic gene polymorphisms, including the *Dopamine transporter 1 (DAT1).*	1999	Israel	The population was divided according to ethnicity: MDD = 102 Ashkenazi *n* = 63 (61.8%) Non-Ashkenazi *n* = 39 (28.2%) Control = 172 Ashkenazi *n* = 112 (65.1%) Non-Ashkenazi *n* = 60 (34.8%)	No statistical differences existed between MDD patients and healthy controls in regard to the researched polymorphisms, including the *SLC6A3 3′UTR VNTR gene* variant (*p* > 0.003).	*p* > 0.003	MDD—Ashkenazi LL (10R/10R) = 23.8% (*n* = 15) LS (10R/9R) = 60.3% (*n* = 38) SS (9R/9R) = 12.7% (*n* = 8) Other genotypes ^2^ = 3.2% (*n* = 2) Control—Ashkenazi ^1^ LL (10R/10R) = 39.3% (*n* = 44) LS (10R/9R) = 42.9% (*n* = 48) SS (9R/9R) = 11.6% (*n* = 13) Other genotypes ^2^ = 5.4% (*n* = 6) MDD—non-Ashkenazi LL (10R/10R) = 30.8% (*n* = 12) LS (10R/9R) = 43.6% (*n* = 17) SS (9R/9R) = 12.8% (*n* = 5) Other genotypes ^2^ = 12.8% (*n* = 5) Control—non-Ashkenazi ^1^ LL (10R/10R) = 45.0% (*n* = 27) LS (10R/9R) = 28.3% (*n* = 17) SS (9R/9R) = 18.3% (*n* = 11) Other genotypes ^2^ = 6.7% (*n* = 4)
Kirchheine et al. ^3^ [19]	“*A 40-basepair VNTR polymorphism in the dopamine transporter (DAT1)* gene and the rapid response to antidepressant treatment”	This prospective cohort study aimed to analyze *DAT1 40 bp VNTR* genetic variant’s influence on the antidepressant response.	2007	Germany	*n* = 190 F = 123 (64.7%) M = 67 (35.3%)	The *SLC6A3 3′UTR VNTR* polymorphism influenced the rapid response to antidepressant therapy. Compared to the LL (10R/10R) carriers, the LS (10R/9R) carriers had a 1.6 odds ratio (OR), and the SS (9R/9R) carriers had a 6.0 OR for no or poor response (*p* = 0.016).	*DAT1 VNTR* polymorphism correlated with rapid response to antidepressant therapy (*p* = 0.016).	LL (10R/10R) = 54% (*n* = 103) LS (10R/9R) = 37% (*n* = 70) SS (9R/9R) = 43% (*n* = 16)
Lavretsky et al. [20]	“The effects of the dopamine and serotonin transporter polymorphisms on clinical features and treatment response in geriatric depression: A pilot study”	Analyze the clinical association between the dopamine and serotonin transporter polymorphism, including the *SLC6A3 3′UTR VNTR* (rs28363170) genetic variant, in late-life depression and preferential treatment response to the methylphenidate–citalopram treatment combination.	2008	USA	MDD *n* = 15 F = 9 (60%) M = 6 (40%)	Individuals with the LL (10R/10R) genotype may be associated with a late-life depression endophenotype, with executive dysfunction that preferentially responds to methylphenidate combined with a selective serotonin reuptake inhibitor (SSRI) to improve mood and cognition.	LL (10R/10R) genotype x decline in depression-severity-based HDRS score reduction (*p* = 0.01).	MDD LL (10R/10R) = 33.3% (*n* = 5) Other genotypes = 66.7% (*n* = 10)
							LL (10R/10R) genotype x methylphenidate combined with citalopram = greater reduction in depression severity throughout treatment (*p* = 0.049).	
Huang et al. [21]	“Association study of the dopamine transporter gene with personality traits and major depressive disorder in the Han Chinese population”	Analyze 17 polymorphisms of the dopamine transporter gene (DAT1) to verify their association with MDD and whether they influence personality traits in MDD patients.	2011	China	MDD *n* = 582 F = 334 (57.4%) M = 248 (42.6%) Control *n* = 435 F = 210 (48.3%) M = 225 (51.7%)	No associations were found between the SLC6A3 (DAT1) 3′UTR VNTR variant’s allelic frequency and MDD (control vs. case group) (*p* = 0.986).	SLC6A3 (DAT1) 3′UTR VNTR variant’s genotypic distributions and allele frequencies did not vary between MDD patients and healthy controls in a Han Chinese population (*p* = 0.986).	MDD LL (10R/10R) = 80.8% (*n* = 470) LS (10R/9R) = 12.4% (*n* = 72) SS (9R/9R) = 0.3% (*n* = 2) Other genotypes = 6.5% (*n* = 38) Control LL (10R/10R) = 80.5% (*n* = 350) LS (10R/9R) = 12.9% (*n* = 56) SS (9R/9R) = 0.5% (*n* = 2) Other genotypes = 6.2% (*n* = 27)
Rafikova et al. [22]	“Influence of Polymorphic Gene Variants of the Dopaminergic System on the Risk of Disorders with Depressive Symptoms”	Identify genetic risk factors for depressive episodes, recurrent depression, and mixed anxiety–depressive disorder. The studied genetic polymorphisms were for *SLC6A3 (DAT1) 40 bp VNTR*, *DRD2 rs1800497*, *DRD4 120 bp VNTR e 48 bp VNTR*, and *COMT rs4680.*	2021	Russia	1st Group— Patients with depressive episodes *n* = 108 F = 63 (58.3%) M = 45 (41.7%)	*SLC6A3 (DAT1) 3′UTR VNTR’s* allelic distribution was statistically significant in the mixed-anxiety-and-depressive-disorder group. The short allele (8R or 9R) was found more in the control group (*p* = 0.005) and was also statistically significant in genotypic distribution (*p* = 0.025).	Genotypic distribution and the depressive episode (*p* = 0.355).	Depressive episode LL = 56.5% (*n* = 61) LS = 38% (*n* = 41) SS = 5.6% (*n* = 6)
					2nd Group—Patients with recurrent depression *n* = 149 F = 101 (67.8%) M = 49 (32.2%)		Genotypic distribution and the risk of depressive episode and recurrent depression (*p* = 0.199).	Recurrent depression LL = 57.0% (*n* = 85) LS = 39.6% (*n* = 59) SS = 3.4% (*n* = 5)
					3rd Group—Mixed anxiety and depressive disorder *n* = 100 F = 52 (52%) M = 48 (48%)		Genotypic distribution and mixed anxiety and depressive disorder (*p* = 0.025).	Mixed anxiety and depressive disorder LL = 52.0% (*n* = 52) LS = 39% (*n* = 39) SS = 9% (*n* = 9)
					Control Group *n* = 163 F = 101 (62%) M = 62 (38%)			Control LL = 69.3% (*n* = 113) LS = 28.2% (*n* = 46) SS = 2.5% (*n* = 4)
Rafikova et al. [14]	“Common and Specific Genetic Risk Factors for Three Disorders with Depressive Symptoms”	Identify genetic risk factors for depressive episodes, recurrent depression, and mixed anxiety–depressive disorder. The studied genetic polymorphisms were for *SLC6A3/DAT1 (locus 40 bp VNTR), DRD2 (locus rs1800497), DRD4 (loci 120 bp VNTR and 48 bp VNTR), COMT (locus rs4680), SLC6A4/5HTT (loci 5-HTTLPR + rs25531 and Stin2), HTR1A (locus rs6295), HTR2A (locus rs6311), HTR1B (locus rs6296),* and *OXTR (locus rs53576).*	2022	Russia	1st Group— Patients with Depressive episode *n* = 106 F = 62 (58.5%) M = 44 (41.5%)	*SLC6A3 (DAT1) 3′UTR VNTR* polymorphism was associated with all depressive episode disorders (*p* = 0.032), recurrent depression (*p* = 0.005), and mixed anxiety and depressive disorder (*p* = 0.005). The SS genotype seems to be a risk factor for mixed anxiety and depressive disorder (OR = 3.93) but not for depression severity.	*SLC6A3 (DAT1) 3′UTR* VNTR’s LL (10R/10R) genotype correlated with the risk of a depressive episode (*p* = 0.032).	Depressive episode ICD F32.1 LL = 55.7% (*n* = 59) LS = 38.6% (*n* = 41) SS = 5.7% (*n* = 6)
					2nd Group— Patients with Recurrent depression *n* = 149 F = 101 (67.8%) M = 48 (32.2%)		*SLC6A3 (DAT1) 3′UTR VNTR’s* LL (10R/10R) genotype correlated with the risk of recurrent depression (*p* = 0.025).	Recurrent depression LL = 57.0% (*n* = 85) LS = 39.6% (*n* = 59) SS = 3.4% (*n* = 5)
					3rd Group—Patients with mixed anxiety and depressive disorder *n* = 97 F = 49 (50.5%) M = 48 (49.5%)		*SLC6A3 (DAT1) 3′UTR VNTR’s* LL (10R/10R) genotype correlated with the risk of mixed anxiety and depressive disorder (*p* = 0.005).	Mixed anxiety and depressive disorder LL = 50.5% (*n* = 49) LS = 40.2% (*n* = 39) SS = 9.3% (*n* = 9)
					Control Group *n* = 163 F = 101 (62%) M = 62 (38%)			Control LL = 69.3% (*n* = 113) LS = 28.2% (*n* = 46) SS = 2.5% (*n* = 4)

^1^ The total sum of the controls (Ashkenazi and non-Ashkenazi) does not agree with the patients’ genotypic distribution. Both ethnic groups are missing an individual; however, no further explanation for their absence is given in the article. ^2^ Frisch et al.’s [18] article was based on the Vandenbergh et al. laboratory methodology. “Other genotypes” include genotypes with alleles 3-to-11 repetitions, excluding those with 9/10 repetitions. ^3^ A patient in Kirchheine et al.’s [19] study had an 8R/10R genotype and was excluded from the genotype frequency calculation.

## Data Availability

No new data were created in this study.

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
