# Peer review of "The *3′UTR VNTR SLC6A3* Genetic Variant and Major Depressive Disorder: A Systematic Review"

_biomedicines, 2023, doi:10.3390/biomedicines11082270_

Round 1

Reviewer 1 Report

My suggestions:

1. I would add a figure on the location of rs28363170, and the differences of SS, 9R/9R, and S/9R genotypes.

2. I would add another figure, which shows, how rs28363170 could impact the depression-related pathways. 

3. Figure 2 should be uploaded in better resolution and bigger font size.

4. In the discussion, I would add a table on SLC6A3 variants, which may impact MDD or other neurological disease development (either intronic or coding).

Author Response

We appreciate the suggestions made and addressed each point raised by the reviewers. We believe that these suggestions increased the overall quality of the submitted manuscript.

Therefore, we are resubmitting our revised systematic review entitled "The 3'UTR VNTR SLC6A3 genetic variant and Major Depressive Disorder: a systematic review.” Changes in the manuscript and our answers to the reviewers’ comments are blue.

All authors are aware of the resubmission and agree with the responses to the reviewers provided below.

____________________________

Reviewer 1

Before: We have some pictures in these answers. Please get the file with the Rebuttal

Comments and Suggestions for Authors

  1. I would add a figure on the location of rs28363170, and the differences of SS, 9R/9R, and S/9R genotypes.

Answer:  We agree and added an illustrative figure of rs28363170 below for evaluation and in the paper’s Introduction. As for the differences between SS, 9R/9R, and S/9R genotypes nomenclature, some authors classify by the specific number of repetitions or by grouping these repetitions - long (≥10) as L and short (<10) as S alleles. Nonetheless, this variant occurs most often with 9 (9R) or 10 (10R) tandem repeat units - the most studied alleles.

Figure 1: The structure of the SLC6A3 3'UTR VNTR (rs28363170) genetic variant's main alleles. (See the file)

  1. I would add another figure, which shows, how rs28363170 could impact the depression-related pathways. 

Answer:  Is the figure below what you’re suggesting?  We agree and added an illustrative figure of rs28363170’s impact on the depression-related pathways below for evaluation and in the Introduction.

Figure 2: A schematic drawing of the SLC6A3 3'UTR VNTR (rs28363170) genetic variant influences dopamine levels at the dopaminergic neuron's postsynaptic and presynaptic regions. (See the file)

  1. Figure 2 should be uploaded in better resolution and bigger font size.

Answer:  We agree and uploaded it with a better resolution and bigger font size, as seen below.

  1. In the discussion, I would add a table on SLC6A3 variants, which may impact MDD or other neurological disease development (either intronic or coding).

Answer:  We do not believe an additional table with this information is necessary as it might distract from the SLC6A3 (DAT1) 3'UTR VNTR 70 (rs28363170) variant effect on MDD, the focus of this systematic review.

Reviewer 2 Report

This article represents systematic review on the role of SLC6A3 (DAT1) 3'UTR VNTR 70 (rs28363170) genetic variant in MDD. This is an interesting topic. The authors have found 6 studies, 4 addressed the association between this variant and the presence of MDD; and two have focused on the treatment response to antidepressants. Due to small number of studies, heterogeneity of populations, small sample sizes and discrepant results, the findings are very preliminary and questinable.

 Some suggestions are provided below:

Introduction

„the total number of individuals suffering from depressive symptoms increased by 18.4% between 2005 and 2015. This upsurge probably is due to global population growth and the expansion of age groups with this disorder“- this is unclear, please, clarify

„MDD patients' central nervous system negatively regulates the levels of neurotransmitters“ – MDD does not regulate brain, changes in the brain eventually produce symptoms, please, correct

Materials and methods

Please, mention also exclusion criteria- „However, articles that presented incomplete data,

such as methodology or statistical data, systematic reviews or meta-analyses, and abstracts, were excluded“ –are these exclusion criteria? Anything else?

What was the time frame for studies?

Results

Table 1. Kirchheine, J., et al. [20] – what was the criteria for rapid response, and to which antidepressants?

Rafikov E. I. et al 2021. and Rafikov E. I. et al 2022. look very similar, the control group and recurrent depression group appear to be the same (regarding the number of participants and genotyping frequencies), other two groups look almost the same, please, clarify and provide more details

Discussion

Major Depressive Disorder (MDD) can be caused by a dysfunction of the dopaminergic reward network that connects the ventral striatum to the orbitofrontal and medial prefrontal cortices- please, provide references, and describe in more details

Please, provide information how, in general, the polymorphisms of DAT gene may associate with antidepressant response, given that 1) antidepressants affect predominantly serotonergic and noradrenergic transmission, and 2) DAT density is very low in prefrontal cortex, but more abundant in limbic system

Please, provide possible explanations for the discrepant findings across studies, and also why would SLC6A3 3'UTR VNTR variant affect suicidallity, what is the possible link between dopaminergic system (especially DAT) and suicidallity in MDD

The biggest limitation is small sample size for genetic studies in all studies, except maybe for Huang, Chang-Chih et al, 2011

Please, highlight the need for future studies regarding genes for dopamine system and treatment response for MDD

Conclusion

"in which evidence indicating that S (9R) allele presence might be a risk factor for developing MDD and L (10R) allele presence a protective factor"-this is based only on two similar studies, and one very small study, while two other studies are negative. So, such evidence is at least very preliminary. The same refers to the abstract. The evidence is very inconclusive.

Author Response

We appreciate the suggestions made and addressed each point raised by the reviewers. We believe that these suggestions increased the overall quality of the submitted manuscript.

Therefore, we are resubmitting our revised systematic review entitled "The 3'UTR VNTR SLC6A3 genetic variant and Major Depressive Disorder: a systematic review.” Changes in the manuscript and our answers to the reviewers’ comments are blue. (Please se the file)

All authors are aware of the resubmission and agree with the responses to the reviewers provided below.

____________________________

Reviewer 2

Comments and Suggestions for Authors

This article represents systematic review on the role of SLC6A3 (DAT1) 3'UTR VNTR 70 (rs28363170) genetic variant in MDD. This is an interesting topic. The authors have found 6 studies, 4 addressed the association between this variant and the presence of MDD; and two have focused on the treatment response to antidepressants. Due to small number of studies, heterogeneity of populations, small sample sizes and discrepant results, the findings are very preliminary and questinable.

 Some suggestions are provided below:

Introduction

  • „the total number of individuals suffering from depressive symptoms increased by 18.4% between 2005 and 2015. This upsurge probably is due to global population growth and the expansion of age groups with this disorder“- this is unclear, please, clarify

Answer:  We decided to remove the phrase. Regardless, it means that as the population grows, so does the percentage in each age group. For instance, the total number of individuals in 10% of thousand is smaller than 10% of a million.

  • „MDD patients' central nervous system negatively regulates the levels of neurotransmitters“ – MDD does not regulate brain, changes in the brain eventually produce symptoms, please, correct

Answer:  I think there is a misunderstanding. The central nervous system (CNS) of MDD patients (MDD patients' central nervous system) is what negatively regulates certain neurotransmitter levels by reducing their activation [10], not MDD. This negative regulation can happen by many mechanisms, including epigenetics, that will eventually produce MDD symptoms.

  1. Bakalov D, Hadjiolova R, Pechlivanova D. Pathophysiology of Depression and Novel Sources of Phytochemicals for its Treatment-A Systematic Review. Acta Medica Bulg. 2020;47(4):69–74.

For instance, Suh et al.’s systematic review and meta-analysis analyzed neurobiological differences between MDD (medicated and medication-naïve) patients (n = 1073) and healthy controls (n = 936) and found changes in anatomical structures that regulate mood, reward-guided behavior, and impulse control. We are not stating that MDD regulates these patients’ brains, but that their CNS is downregulating the levels and, therefore, activities of different neurotransmitters, such as serotonin (5-HT), norepinephrine (NE), and dopamine (DA).

Suh, J.S.; Schneider, M.A.; Minuzzi, L.; MacQueen, G.M.; Strother, S.C.; Kennedy, S.H.; Frey, B.N. Cortical thickness in major depressive disorder: A systematic review and meta-analysis. Prog. Neuro-Psychopharmacol. Biol. Psychiatry 2019, 88, 287–302.

Materials and methods

  • Please, mention also exclusion criteria- „However, articles that presented incomplete data, such as methodology or statistical data, systematic reviews or meta-analyses, and abstracts, were excluded“ –are these exclusion criteria? Anything else?

Answer: Yes, these exclusion criteria together with anything that didn’t follow the PECOS inclusion criteria:

“(1) population: research participants with Major Depressive Disorder (MDD); (2) Exposure: SLC6A3 3'UTR VNTR gene variant (rs28363170); (3) Comparison: the SLC6A3 3'UTR VNTR genetic variant's SS (9R/9R) genotypic frequency, S (9R) allelic frequency, or both; (4) Outcome: the SLC6A3 3'UTR VNTR genetic variant's SS (9R/9R)/S (9R) frequency fluctuation in different populations; (5) study type: observational and intervention.”

In other words, articles were excluded if none of the research participants analyzed had Major Depressive Disorder (MDD) or if the  SLC6A3 3'UTR VNTR gene variant (rs28363170) was not analyzed, or if its genotypic/allelic frequency was not presented in any form for the MDD participants, or if study type was not observational and intervention.

  • What was the time frame for studies?

Answer:  The search was not restricted by a time frame as very few works analyzed this variant in MDD patients. This topic is a literature background check for our analysis of this SLC6A3 3'UTR VNTR gene variant (rs28363170) in a Brazilian sample population with Major Depressive Disorder (MDD) and to help answer some of our benchwork questions. However, we found few studies, so we decided not to time restrain the search.

Results

  • Table 1. Kirchheine, J., et al. [20] – what was the criteria for rapid response, and to which antidepressants?

Answer:  The criterion Kirchheine et al. (20) used to verify the rapid response to antidepressants was the reduction in the Hamilton Depression Rating Scale (HDRS) score within the first three weeks after starting antidepressant therapy (their study period). The antidepressants used were mirtazapine, SSRIs (sertraline, citalopram, paroxetine, fluvoxamine, fluoxetine, and venlafaxine), tricyclic antidepressants (amitriptyline, clomipramine, doxepin, and trimipramine) and others (reboxetine, nefazodone, moclobemide, mianserin, and the herb Hypericum perforatum, also known as St. John's wort). Research participants were treated with medication to manage symptoms and underwent individual and group psychotherapy and psychotherapeutic interview.

  • Rafikov E. I. et al 2021. and Rafikov E. I. et al 2022. look very similar, the control group and recurrent depression group appear to be the same (regarding the number of participants and genotyping frequencies), other two groups look almost the same, please, clarify and provide more details

Answer: Thank you for noticing it in time. Rafikov et al.'s first published study (2021) investigated the association of the dopaminergic system's genetic polymorphisms with three types of severe depressive disorders: Depressive Episode (DE), Recurrent Depression (DR), and Mixed Anxiety and Depression Disorder (MADD). In contrast, their 2022 study pooled the genotyping results for the loci of the dopaminergic, serotonergic, and endocrine systems and tested their association with the same depressive disorders but using various inheritance models. They also analyzed whether the studied loci correlated to the severity of the depressive and anxious symptoms.  We have rewritten phrases in the discussion to clarify this difference to the reader (see manuscript).

Discussion

  • Major Depressive Disorder (MDD) can be caused by a dysfunction of the dopaminergic reward network that connects the ventral striatum to the orbitofrontal and medial prefrontal cortices- please, provide references, and describe in more details.

Answer: Thank you for pointing it out as forgot to add to the phrase although they were cited in the paragraph. The phrase’s references are listed below and have been added to the end of the sentence in the manuscript.

HELLWIG, Sabine et al. Antidepressant treatment effects on dopamine transporter availability in patients with major depression: a prospective 123 I-FP-CIT SPECT imaging genetic study. Journal of Neural Transmission, v. 125, p. 995-1005, 2018.

SAMBATARO, Fabio et al. A variable number of tandem repeats in the 3′‐untranslated region of the dopamine transporter modulates striatal function during working memory updating across the adult age span. European Journal of Neuroscience, v. 42, n. 3, p. 1912-1918, 2015.

Rafikova EI, Shibalev D V, Shadrina MI, Slominsky PA, Guekht AB, Ryskov AP, et al. Influence of Polymorphic Gene Variants of the Dopaminergic System on the Risk of Disorders with Depressive Symptoms. Russ J Genet [Internet]. 2021;57(8):942–8. Available from

Thanks to your questions, we decided to alter the introductory paragraph in the discussion to expand background information, as written below:

“Major Depressive Disorder (MDD) can be caused by a dysfunction of the dopaminergic reward network that connects the ventral striatum to the orbitofrontal and medial prefrontal cortices [22,23,24, GRACE, 2016]. One of the mechanisms responsible for regulating the dopaminergic system is the dopamine transporters (DAT), which, together with other mechanisms, help modulate extracellular striatal dopamine (DA) concentrations in the synaptic cleft in brain regions, mainly in cortical and subcortical regions (FELTEN et al., 2011). Located on the membrane of presynaptic terminals, DAT is responsible for regulating the intensity and duration of dopaminergic transmission in the synaptic cleft, the reuptake of dopamine from presynaptic cells in the striatum and midbrain, in addition to interfering with the clearance of dopamine in the extracellular striatum region (PIZZAGALLI et al., 2019). Reducing dopamine levels diminishes striatal DAT's density and functionality - a compensatory downregulation feedback mechanism to adjust the reduced DA concentration) (PIZZAGALLI et al., 2019). The SLC6A3 (DAT1) 3'UTR VNTR (rs28363170) polymorphism alters DAT expression, with SS (9R/9R) genotype carriers tending to be associated with a more severe depression course -(Rafikov E. I. et al 2021).”

  • Please, provide information how, in general, the polymorphisms of DAT gene may associate with antidepressant response, given that 1) antidepressants affect predominantly serotonergic and noradrenergic transmission, and 2) DAT density is very low in prefrontal cortex, but more abundant in limbic system

Answer: 1) Grace's 2016 review confirmed a correlation between the dysfunction of the dopaminergic system and the pathophysiology of depression and schizophrenia, with some depressive symptoms characterized as an effect of dopaminergic dysfunction, such as anhedonia and lack of motivation (GRACE, 2016). One of those responsible for regulating the dopaminergic system is the dopamine transporters (DAT), which, together with other mechanisms, help modulate dopamine concentrations in the synaptic cleft in brain regions, mainly in cortical and subcortical regions (FELTEN et al., 2011). Located on the membrane of presynaptic terminals, DAT is responsible for regulating the intensity and duration of dopaminergic transmission in the synaptic cleft, the reuptake of dopamine from presynaptic cells in the striatum and midbrain, in addition to interfering with the clearance of dopamine in the extracellular striatum region (PIZZAGALLI et al., 2019). A reduction in dopamine synthesis diminishes the DAT's density and functionality (a compensatory feedback mechanism to adjust to this reduction by downregulation of DAT synthesis), which is a mechanism to trigger depression (PIZZAGALLI et al., 2019). Many antidepressant treatments affect DAT availability by acting directly on the dopaminergic pathway (HELLWIG et al., 2018). An example is desipramine, which raises dopamine concentration in the prefrontal cortex, reducing the chances of developing depression (LEGGIO et al., 2013). In this way, polymorphisms that affect dopamine transporter (DAT1) transcription/expression will also interfere with this feedback mechanism and might also affect/promote the development of neuropsychiatric disorders such as depression. The SLC6A3 3'UTR VNTR variant impacts exactly DAT expression, with the 9R allele conferring a lower DAT expression than the 10R allele. Therefore, understanding the mechanisms of pharmacogenetics is extremely important to understand the disease susceptibility/development and promote the development or better adequation of treatment for the patient.

GRACE, A. A. Dysregulation of the dopamine system in the pathophysiology of schizophrenia and depression. Nature Reviews Neuroscience, v. 17, n. 8, p. 524–532, 2016.

FELTEN, A. et al. Genetically determined dopamine availability predicts disposition for depression. Brain and Behavior, v. 1, n. 2, p. 109–118, 2011.

PIZZAGALLI, D. A. et al. Assessment of Striatal Dopamine Transporter Binding in Individuals with Major Depressive Disorder: In Vivo Positron Emission Tomography and Postmortem Evidence. JAMA Psychiatry, v. 76, n. 8, p. 854–861, 2019.

LEGGIO, G. M. et al. Dopamine D3 receptor as a new pharmacological target for the treatment of depression. European Journal of Pharmacology, v. 719, n. 1–3, p. 25–33, 2013.

HELLWIG, S. et al. Antidepressant treatment effects on dopamine transporter availability in patients with major depression: a prospective 123I-FP-CIT SPECT imaging genetic study. Journal of Neural Transmission, v. 125, n. 6, p. 995–1005, 2018.

2) We need further clarification on the question in this part. Nevertheless, DAT is also responsible for transporting dopamine inside or outside the neuron to control synapse levels. The most significant amount of dopamine removal from the synapse performed by neuronal DAT is in the dorsal striatum region and the nucleus accumbens. (BAHI; DREYER, 2019). Antidepressants, such as desipramine, in rats' prefrontal cortex inhibited noradrenaline reuptake while increasing dopamine extracellular concentrations. Fluoxetine, a selective serotonin reuptake inhibitor, can also increase dopamine extracellular concentrations in the prefrontal cortex by a non-serotonin-dependent mechanism; and lastly, bupropion, a selective dopamine reuptake inhibitor, conducts a blockade of DATs (DAILLY et al., 2004). Therefore, understanding a disease's pathophysiology and associating it with pharmacogenomics becomes essential to promote patients' quality of life.

BAHI, A.; DREYER, J. L. Dopamine transporter (DAT) knockdown in the nucleus accumbens improves anxiety- and depression-related behaviors in adult mice. Behavioural Brain Research, v. 359, n. October 2018, p. 104–115, 2019.

DAILLY, E. et al. Dopamine, depression and antidepressants. Fundamental and Clinical Pharmacology, v. 18, n. 6, p. 601–607, 2004.

  • Please, provide possible explanations for the discrepant findings across studies, and also why would SLC6A3 3'UTR VNTR variant affect suicidallity, what is the possible link between dopaminergic system (especially DAT) and suicidallity in MDD

Answer: The probability of the SLC6A3 3'UTR VNTR variant affecting suicidal behavior, depression, and violence is due to its influence com DAT transcription, that is, its impact on dopamine levels at neuron's postsynaptic and presynaptic regions, therefore, influence on serotonergic or dopaminergic pathways (RYDING; LINDSTRÖM; TRÄSKMAN-BENDZ, 2008). Typically, patients with depression and high suicidality had reduced striatal DAT concentration compared to individuals with a low probability of committing suicide, presumably due to compensatory down-regulation secondary to dopamine signaling within the mesolimbic pathways (DUVAL et al., 2022). Nonetheless, only one study analyzed and correlated suicide with the SLC6A3 3'UTR VNTR polymorphism (RAFIKOVA et al., 2021). Thus, more multicentric studies are needed, as the genetic heterogeneity in different populations will allow for a greater understanding of suicide, prevention, and treatment.

DUVAL, F. et al. Dopamine Function and Hypothalamic-Pituitary-Thyroid Axis Activity in Major Depressed Patients with Suicidal Behavior. Brain Sciences, v. 12, n. 5, 2022.

RAFIKOVA, E. et al. SLC6A3 (DAT1) as a Novel Candidate Biomarker Gene for Suicidal Behavior. GENES, v. 12, n. 6, 2021.+++

RYDING, E.; LINDSTRÖM, M.; TRÄSKMAN-BENDZ, L. The role of dopamine and serotonin in suicidal behaviour and aggression. Progress in Brain Research, v. 172, n. 08, p. 307–315, 2008.

  • The biggest limitation is small sample size for genetic studies in all studies, except maybe for Huang, Chang-Chih et al, 2011

Answer:  Yes, our most significant limitations were the small sample size and the small number of studies that passed our PECOS inclusion criteria. Even completely opening the time frame didn't help increase the number of studies.

  • Please, highlight the need for future studies regarding genes for dopamine system and treatment response for MDD

Answer:  We highlighted the need for further research, even in opening it up to more dopamine transporter variants, in the conclusion:

"Understanding the SLC6A3 gene, its polymorphisms, and its mechanisms' role in MDD etiology might contribute to the establishment of new, more modern forms of early diagnosis and increases patients' quality of life."

Adding all the genes part of the dopamine signaling pathway system might broaden the scope too much. Not explicitly stated does not convey that there is no necessity to study the entire system; just the role of the SLC6A3 gene, its polymorphisms, and its mechanisms' role in MDD is still lacking.

Conclusion

  • "in which evidence indicating that S (9R) allele presence might be a risk factor for developing MDD and L (10R) allele presence a protective factor"-this is based only on two similar studies, and one very small study, while two other studies are negative. So, such evidence is at least very preliminary. The same refers to the abstract. The evidence is very inconclusive.

Answer:  We agree and modified the conclusion to:

"This systematic review found preliminary agreement regarding the SLC6A3 (DAT1) 3'UTR VNTR (rs28363170) polymorphism association with MDD is inconclusive. Two similar studies with the same population sample and a small sample study indicate that S (9R) allele presence may be a risk factor for developing MDD and L (10R) allele presence a protective factor. In contrast, the two remaining studies indicated that this polymorphism does not influence MDD.

The limitations of this study are due to the few articles that met the eligibility criteria for this review, and consequently, very few populations with different ethnicities were analyzed. Therefore, further studies with a significant sample size are needed to assess the behavior of this genetic variant's genotypic/allelic frequency distribution in diverse populations and help understand this polymorphism's role in MDD conclusively.

Understanding the SLC6A3 gene, its polymorphisms, and its mechanisms' role in MDD etiology might contribute to the establishment of new, more modern forms of early diagnosis and increases patients' quality of life.”

Round 2

Reviewer 1 Report

Authors fulfilled my suggestions, thank you.

Reviewer 2 Report

The authors did a good work accepting all suggestions and providing some additional clarifications